# Doxorubicin and Cisplatin Modulate miR-21, miR-106, miR-126, miR-155 and miR-199 Levels in MCF7, MDA-MB-231 and SK-BR-3 Cells That Makes Them Potential Elements of the DNA-Damaging Drug Treatment Response Monitoring in Breast Cancer Cells—A Preliminary Study

**DOI:** 10.3390/genes14030702

**Published:** 2023-03-12

**Authors:** Anna Mizielska, Iga Dziechciowska, Radosław Szczepański, Małgorzata Cisek, Małgorzata Dąbrowska, Jan Ślężak, Izabela Kosmalska, Marta Rymarczyk, Klaudia Wilkowska, Barbara Jacczak, Ewa Totoń, Natalia Lisiak, Przemysław Kopczyński, Błażej Rubiś

**Affiliations:** 1Department of Clinical Chemistry and Molecular Diagnostics, Poznan University of Medical Sciences, Rokietnicka 3, 60-806 Poznan, Poland; 2Centre for Orthodontic Mini-Implants, Department and Clinic of Maxillofacial Orthopedics and Orthodontics, Poznan University of Medical Sciences, Bukowska 70, 60-812 Poznan, Poland

**Keywords:** miRNA, breast cancer, diagnostics, therapy response

## Abstract

One of the most innovative medical trends is personalized therapy, based on simple and reproducible methods that detect unique features of cancer cells. One of the good prognostic and diagnostic markers may be the miRNA family. Our work aimed to evaluate changes in selected miRNA levels in various breast cancer cell lines (MCF7, MDA-MB-231, SK-BR-3) treated with doxorubicin or cisplatin. The selection was based on literature data regarding the most commonly altered miRNAs in breast cancer (21-3p, 21-5p, 106a-5p, 126-3p, 126-5p, 155-3p, 155-5p, 199b-3p, 199b-5p, 335-3p, 335-5p). qPCR assessment revealed significant differences in the basal levels of some miRNAs in respective cell lines, with the most striking difference in miR-106a-5p, miR-335-5p and miR-335-3p—all of them were lowest in MCF7, while miR-153p was not detected in SK-BR-3. Additionally, different alterations of selected miRNAs were observed depending on the cell line and the drug. However, regardless of these variables, 21-3p/-5p, 106a, 126-3p, 155-3p and 199b-3p miRNAs were shown to respond either to doxorubicin or to cisplatin treatment. These miRNAs seem to be good candidates for markers of breast cancer cell response to doxorubicin or cisplatin. Especially since some earlier reports suggested their role in affecting pathways and expression of genes associated with the DNA-damage response. However, it must be emphasized that the preliminary study shows effects that may be highly related to the applied drug itself and its concentration. Thus, further examination, including human samples, is required.

## 1. Introduction

Breast cancer is still among women’s most common leading causes of death worldwide [1]. The main challenge is an early diagnosis as well as personalized therapy adjustment. They both require the identification of specific qualitative and quantitative assessments of reliable parameters that could be used as markers and therapy targets, respectively. The main obstacle to therapy efficacy is the resistance of cancer cells to drugs. This is driven by different pathways, including individual genetic characteristics and multi-drug resistance, cell death inhibition (apoptosis suppression), altered drug metabolism, epigenetics, enhanced DNA repair and gene amplification [2]. As reported below (Table 1), all these processes can be controlled by the specific miRNA-controlled expression of certain genes. Another critical characteristic of breast cancer cells (apart from immortality) is the ability to metastasize associated with altered adhesion, migration and invasion. It is mainly related to signaling pathways controlled by different genes or their phosphorylated/dephosphorylated expression products.

Moreover, miRNAs can control genes that regulate adhesion molecules and cell–cell interactions (Table 1). Consequently, it is difficult to identify a specific signature of cancer cells with so many variables. Nevertheless, epigenetic factors can significantly modulate these processes, and microRNA profiling seems to be a promising strategy in diagnostics and therapy response monitoring. Among numerous studies performed so far, some particular miRNAs can be perceived as suitable candidates, especially since most of them are commonly associated with different cancer types, including breast cancer. The idea of miRNA functioning as early biomarkers for the evaluation of drug efficacy and drug safety was proposed not only in cancer but also in other diseases, including multiple sclerosis [3], bipolar disorder [4], diabetes [5] and others [6]. Thus miRNA profiling, supported by bioinformatic analysis, is perceived as a specific and sensitive biomarker for evaluating drug efficacy/resistance and drug safety in patients [7]. Specifically, it was shown that blood-borne miRNA profiles monitored over time have the potential to predict complete pathological responses in breast cancer [8]. Altogether, miRNA assessment shows some potential as a marker in pharmacogenomics (as modulators of pharmacology-related genes). It can be evaluated using low-invasive methods providing high specificity and sensitivity. However, since one miRNA can target different mRNA, a comprehensive profiling study must be performed to obtain an informative and valuable pattern.genes-14-00702-t001_Table 1Table 1Contribution of selected miRNAs to breast cancer metabolism.miRNAEffect and Pathways Mediated in Breast CancerRef.miRNA-21(-3p and -5p)tumor growth, cancer cells proliferation, metastasis, invasion, sensitivity to chemotherapy, modulation of cancer-related gene expression[9,10]miRNA-106a-5pcancer cell proliferation, colony-forming capacity, migration, invasion, breast cancer cell apoptosis and sensitivity to cisplatin, DNA damage response, suppression of the ATM-associated pathway[11,12]miRNA-126(-3p and -5p)cancer cell migration, tumor growth, proliferation, invasion and angiogenesis of triple-negative breast cancer cells[13,14,15]miRNA-155(-3p and -5p)inflammation, apoptosis, adhesion[1,16,17]miRNA-199b(-3p and -5p)cancer aggressiveness, tumor growth, angiogenesis[18,19,20]miRNA-335(-3p and -5p)sensitivity of triple-negative breast cancer cells to paclitaxel, cisplatin and doxorubicin[21]

## 2. Materials and Methods

### 2.1. Cell Culture

Three cell lines representing different molecular subtypes of breast cancer were enrolled in the study, i.e., (i) MCF7 (ER/PR+, HER2low, TP53WT), (ii) MDA MB-231—basal-like subtype, also called triple-negative breast cancer (TNBC; ER/PR-, HER2-, TP53mut), and (iii) SK-BR-3 (ER/PR-, HER2+, TP53mut). MCF7 cell line, in comparison to the MDA-MB-231 cell line, is a poorly aggressive and non-invasive cell line. Overall, it is being considered to have low metastatic potential. SK-BR-3 cells are the least invasive cells out of the three studied, according to previous findings [22]. The MCF7 (HTB-22) and MDA-MB-231 (HTB-26) cells were maintained in RPMI-1640 (Biowest, Nuaillé, France), while SK-BR-3 (HTB-30) cells were cultured in McCoy’s 5A (Biowest, Nuaillé, France) media supplemented with 10% fetal bovine serum (FBS) (Biowest, Nuaillé, France) at 37 °C in an atmosphere of 5% CO_2_ and saturated humidity. All cell lines were obtained from the American Type Culture Collection (ATCC).

### 2.2. Studied Drugs

Both studied drugs belong to the DNA-damaging and proapoptotic agents, and both are ABC family substrates (ABCB1 and ABCC3, respectively) [23]. Doxorubicin is commonly used in breast cancer treatment disruption of topoisomerase-II-mediated DNA repair and generation of free radicals and their damage to cellular membranes, DNA and proteins [24]. The mechanism of resistance to doxorubicin results from a reduction in the ability of the drug to accumulate in the nucleus, decreased DNA damage and suppression of the downstream events that transduce the DNA damage signal to apoptosis [25]. 

In turn, cisplatin effectively blocks breast cancer metastasis and inhibits cancer growth together with paclitaxel in neoadjuvant chemotherapy [26]. Resistance of cancer cells to this drug is associated with decreased drug import, increased drug export, increased drug inactivation by detoxification enzymes, increased DNA damage repair and inactivated cell death signaling [27].

### 2.3. MTT Cell Survival Assay

Cell survival was determined using MTT assay by assessing the sensitivity of cells subjected to drugs, i.e., doxorubicin or cisplatin, as previously described [28]. Drugs selection was based on common use in breast cancer [29]. Briefly, the cells were seeded at a density of 5 × 10^3^ cells per well in 96-well culture plates, incubated overnight to allow for cell attachment, and the next day either DOX was added at a concentration range of 0, 0.05, 0.1, 0.5, 1, 2 or 5 μM or cis-Pt was administered at a concentration range of 0, 1, 2, 5, 10, 20 or 50 μM (DOX was dissolved in water, and cis-Pt in saline). Cells were treated for 24 h and incubated with 10 μL of MTT reagent (5 mg/mL) (Sigma-Aldrich, St. Louis, MO, USA). The cells were incubated at 37 °C for 4 h, followed by the addition of 100 μL of solubilization buffer (10% SDS in 0.01 M HCl). The absorbance was measured in each well with the Microplate Reader Multiskan FC (Thermo Scientific, Waltham, MA, USA) at two wavelengths of 570 and 690 nm. Each experimental point was determined in biological triplicate (each in 6 technical repeats). IC_50_ (half-maximal (50%) inhibitory concentration) values were calculated using CompuSyn (ComboSyn, Inc., Paramus, NJ, USA) (Table 2), and the standard deviation was calculated using Excel software (Microsoft, Syracuse, NY, USA).

### 2.4. miRNA Isolation

Cells were subjected to treatment with DOX (0.1 μM) or cis-Pt (10 μM) for 24 h. Concentration selection resulted from the survival curve obtained in MTT assay and these specific concentrations were chosen as subcytotoxic but known from previous experiments to provoke apoptosis in a longer incubation time (verified by clonogenic assay, data not shown).

miRNAs were isolated using a miRNA Isolation Kit Cell (BioVendor, Czech Rep.) based on optimized silica membrane column according to manufacturers’ instructions. Briefly, dedicated cell lysis buffer (250 μL per sample; 1% β-mercaptoethanol freshly added) was added to cell pellets (1 × 10^6^ cells) followed by vortexing, a brief spin and incubation at 25 °C (room temperature) for 3 min. Next, RCL1 and RCL2 buffers were sequentially added that was accompanied by brief vortexing, spinning and short incubation at 25 °C (room temperature). After centrifugation at 11,000× *g* for 3 min, the clear supernatant was transferred to a new 1.5 mL micro-centrifuge tube, 330 μL of isopropanol was added and after short pulse-vortexing for 10 s, all content was transferred to the MR13 Column, followed by incubation at 25 °C for 2 min. After another centrifugation at 11,000× *g* for 1 min, 500 μL of buffer CRW1 was added to wash the column, followed by another centrifugation at 11,000× *g* for 1 min. After washing the column with 500 μL of Buffer CRW2, the miRNAs were eluted with 30 μL of RNase-Free water (centrifugation at 11,000× *g* for 1 min), assessed using spectrophotometer (Eppendorf Biophotometer Plus Spectrophotometer, Hamburg, Germany) and stored at −80 °C for further study.

### 2.5. qPCR Assessment of miRNAs

Two-Tailed qPCR (BioVendor, Brno, Czech Republic) was performed to assess individual miRNAs according to manufacturers’ instructions. The test is based on primers which consist of two hemiprobes, connected by a folded tether. Complementarity of two hemiprobes provides specific binding and specific cDNA synthesis. The cDNA was obtained using the miR-TT-PRI kit containing set of miRNA-specific primers and the Two-Tailed cDNA Synthesis System using a thermocycler (Eppendorf EP Gradient S Thermocycler, Hamburg, Germany) according to the following protocol: 25 °C for 5 min, 50 °C for 15 min, 85 °C for 5 min followed by cooling at 4 °C. For each sample, 600 ng of total RNA was taken for reverse transcription—altogether, it was a combination of three biological repeats (200 ng of each) in one sample. Next, specific PCR primers (PCR Primer F and PCR Primer R) from the miR-TT-PRI kit were added. The qPCR was performed as follows: 95 °C for 30 s, 95 °C for 5 s, 60 °C for 15 s, 72 °C for 10 s and signal detection. After 40 cycles, melting curve analysis was performed and final, relative expression was evaluated based on Cq (quantification cycle) and thermocycler software (Roche LightCycler 480-II PCR, Basel, Switzerland) as previously described [30]. Quantitative qPCR was done with individual reactions for each miRNA target. Validation was performed against U6 (RNU6-1) expression as previously described [31]. Melting temperature analysis was used (SYBRGreen-based) for verification of specific products detection.

### 2.6. Statistical Analysis

Results were expressed as mean ± SD. All statistical analyses were carried out using GraphPad Prism 5 (GraphPad Software, Sandiego, CA, USA). Differences were assessed for statistical significance using repeated-measures ANOVA, followed by the post-hoc Dunnett’s test method. All experiments were performed in triplicates unless specified otherwise. The threshold for significance was defined as *p* < 0.05 and are indicated by the (*, #, •) symbol for *p* < 0.05. qPCR was performed with pooled samples, as commonly acknowledged [32], giving mean target miRNAs levels.

## 3. Results

### 3.1. Doxorubicin and Cisplatin Cytotoxicity Evaluation

All three cell lines were subjected to different concentrations of studied drugs to find the range of low toxicity concentrations that could be applied to cells during the evaluation of the association between drug treatment and cell response measured by alterations in selected miRNAs levels. The 24 h time interval selection was based on our previous experience and numerous cytotoxicity assays, as well as more mechanistic assessment of apoptotic pathways, induced by the studied drugs. We observed that longer incubation time (48 and 72 h MTT test were performed, showing more than 40% viability decrease in all three cell lines at the lowest concentrations of both studied drugs; data not included) would provoke cytotoxic effect that might not reflect mechanistic aspect of the specificity of the cancer cells response to the treatment. Additionally, according to our previous studies, many genes are regulated within 24 h after DOX or cis-Pt treatment. Similarly, numerous reports indicate 24 h time interval as a sufficient and optimal time to observe significant alterations in miRNA levels after cancer cells treatment (e.g., [33,34,35,36]). For this reason, we were interested in a rapid response that was supposed to be specific. Thus, we decided to verify our hypothesis concerning monitory potential of miRNAs in breast cancer cells in a 24 h time interval that was supposed to provide an early response analysis.

An MTT survival test was involved for DOX (Figure 1A) and cisplatin (Figure 1B) cytotoxicity assessment. Cell treatment was 24 h and, as demonstrated, cytotoxicity effects were concentration dependent. MCF7 and MDA-MB-231 showed a similar survival rate when treated with DOX at 0.05, 0.1, 0.5 or 1 μM (up to ca 20% decrease in survival, relative to control, untreated cells). Simultaneously, treatment of SK-BR-3 cells with the same concentrations of DOX showed significant decrease in survival only at the concentration of 0.5 and 1 μM (up to 20% decrease in survival, relative to control cells). In turn, higher concentration (2 μM) revealed higher resistance of MDA-MB-231 than two other cell lines (survival at 80 vs. 65% in MCF7 and 60% in SK-BR-3), while incubation with 5 μM DOX led to similar survival rate in all three cell lines i.e., 60% (Figure 1A).

The same cell lines were treated with cisplatin at the range of concentrations 1, 2, 5, 10, 20 or 50 μM. When cells were subjected to 1, 2, 5 or 10 μM cisplatin, the survival rate of MCF7 and MDA-MB-231 was reduced by circa 10%. At the same time viability of SK-BR-3 was unchanged. Increasing the concentration of cis-Pt (20 or 50 μM) provoked decrease of all three breast cancer cell lines survival, with a more dominant effect observed in SK-BR-3 (50 vs. 20% decrease at 50 μM) (Figure 1B).

Based on the assessment of toxicity of the drugs, IC_50_ values were calculated (Table 1). Primarily, time-course experiments were performed, but since after longer time intervals (48 or 72 h; data not shown), the compounds appeared highly toxic, we decided to subject cells to 24 h treatment only. Low-cytotoxicity concentrations were selected based on the survival curves (these concentrations, however, are known from previous experiments and literature data to induce apoptosis).

### 3.2. Target miRNAs Selection

First, we used the targetscan algorithm for selection of target miRNAs that were supposed to be evaluated [37]. However, as it gave us enormous, not entirely coherent data, and bearing in mind that miRNAs do not have to be fully complementary to interact with target mRNAs, we performed a selection based on literature data. Additionally, TCGA analysis (GEPIA2 [38], Xena Browser [39] and oncolnc.org (access date: 27 February 2023) [40]) was also performed and discussed below.

Consequently, eleven most commonly breast cancer-associated miRNAs were subjected to identification after doxorubicin (DOX) or cisplatin (cis-Pt) treatment of breast cancer cell lines. Additionally, assessment of a synthetic nonmammalian miR-54-3p was performed as a negative control. The target miRNAs (with short justification) were as follows:

*miRNA-21 (-3p and -5p)*. miR-21-5p was identified as a typical onco-miRNA. When upregulated, it could promote tumor growth, metastasis and invasion and reduce sensitivity to chemotherapy. It modulates the expression of multiple cancer-related target genes and is dysregulated in various tumors [9]. Decreased expression of miR-21 is known to suppress the invasion and proliferation of MCF7 cells [10].

*miRNA-106a-5p*. In human breast cancer, miR-106a expression was found to be significantly upregulated. It enhanced breast cancer cell proliferation, colony-forming capacity, migration and invasion. Additionally, miR-106a overexpression significantly decreased breast cancer cell apoptosis and sensitivity to cisplatin [11]. Upregulation of miRNA-106a modified DNA damage response and led to the suppression of the ATM gene and formation of its protein product at nuclear foci [12].

*miRNA-126 (-3p and -5p)*. Studies suggest that miR-126-3p acts as either a tumor suppressor or an oncogene in different types of cancer. Upregulation of miR-126-5p can inhibit the migration of the MCF7 breast cancer cell line [13]. Furthermore, overexpression of miR-126-3p significantly reduced tumor size [14]. miRNA-126-3p overexpression inhibited the proliferation, migration, invasion and angiogenesis of triple-negative breast cancer cells (MDA-MB-231 and HCC1937) [15].

*miRNA-155 (-3p and -5p)*. miR-155 was shown to play a crucial role in various physiological and pathological processes, including inflammation and cancer. It was found overexpressed in several solid tumors, including breast, colon, cervical and lung cancers [1] and it is supposed to mediate the pathway controlled by caspase-3 [16]. The mechanism of action of this miRNA is based on downregulation of the cell adhesion molecule 1 (CADM1) that functions as a tumor suppressor [17].

*miRNA-199b (-3p and -5p)*. miR-199b-5p was reported to play a critical role in various types of malignancy. There are studies suggesting that miR-199b-5p downregulation is correlated with aggressive clinical characteristics of breast cancer [18,19]. Overexpression of miR-199b-5p inhibited the formation of capillary-like tubular structures and reduced breast tumor growth and angiogenesis in vivo [20]. Downregulation of miR-199b-5p is correlated with poor prognosis for breast cancer patients.

*miRNA-335 (-3p and -5p).* The expression of miR-335 depends on the type of cancer. It is downregulated in breast cancers and upregulated in gallbladder carcinoma, endometrial and gastric cancers. Overexpression of miR-335 increases the sensitivity of triple-negative breast cancer cells to paclitaxel, cisplatin and doxorubicin, and improves the effectiveness of chemotherapy. It is also associated with cisplatin sensitivity in ovarian cancer [21].

To summarize the contribution of selected miRNAs to breast cancer metabolism, the justification was collected in Table 2.

### 3.3. Quantitative Assessment of the Basal Levels of Selected miRNAs

All three cell lines were subjected to quantitative assessment of the basal levels of selected miRNAs using qPCR and relative quantification. All biological experiments were performed in triplicates followed by RT-PCR and qPCR assessment. Consequently, all target miRNAs levels were shown as relative to respective targets in MCF7 cells, used as a calibrator (value “1” assigned to basal level of each miRNA in MCF7). Additionally, the data were divided into three groups i.e., relatively high miRNA levels, low levels and other, relative to results observed in MCF7 cells (selected as reference cell line; Figure 2A,B and Figure 3C, respectively).

As demonstrated, average basal levels of individual miRNAs were different in different cell lines with no specific pattern. All the studied miRNAs (21-3p, 21-5p, 106a-5p, 126-3p, 126-5p, 155-3p, 155-5p, 199b-3p, 199b-5p, 335-3p, 335-5p) were detected in MCF7 and MDA-MB-231, while SK-BR-3 did not show expression of the miR-155-3p (Figure 2A). Reaction designed for detection of a synthetic nonmammalian cel-miR-54-3p was used as a negative control.

### 3.4. Cancer Cells Response to Drugs—Evaluation of the Potential Association between Drug Treatment and miRNA Levels

All three cell lines were subjected to selected miRNAs assessment after DOX (0.1 μM) or cis-Pt (10 μM) 24 h treatment. Some alterations of studied miRNAs were observed with no significant pattern but with some consistency between the two applied DNA-damaging drugs within particular cell lines. We decided to focus on the miRNAs that showed at least a 40% change relative to control samples.

#### 3.4.1. miRNA Alterations in MCF7 Cells

The qPCR showed that DOX treatment of MCF7 cells provoked downregulation of 126-3p, 155-3p, 199b-3p and 335-5p miRNAs (Figure 3). The same cells treated with cisplatin (10 μM) revealed induction of miRNAs 155-3p and 335-5p. At the same time, 126-3p and 199b-3p miRNAs were downregulated by cis-Pt (Figure 3A).

#### 3.4.2. miRNA Alterations in MDA-MB-231 Cells

When MDA-MB-231 cells were treated with DOX, an induction of 21-3p, 155-3p and 199b-5p was observed. At the same time, treatment of these cells with DOX provoked downregulation of 126-3p and 199b-3p (Figure 3B). Incubation of MDA-MB-231 cells with cisplatin caused increased accumulation of 106a-5p and 155-3p miRNAs, while 21-5p, 126-3p, 126-5p and 199-3p were downregulated (Figure 3B).

#### 3.4.3. miRNA Alterations in SK-BR-3

Evaluation of miRNAs alterations in SK-BR-3 cells subjected to DOX treatment showed that cells treated with the drug triggered accumulation of 106a-5p, 155-5p and 199b-3p. At the same time 21-3p, 21-5p, 155-3p, 335-3p and 335-5p were downregulated (Figure 3C). When cells were incubated with cisplatin, upregulation of 106a-5p was observed, while 21-3p, 21-5p, 155-3p, 199-5p, 335-3p and 335-5p were downregulated (Figure 3C).

## 4. Discussion

Epigenetics seems to play a pivotal role in the metabolism of all human cells, including cancer cells. One of the mechanisms involved in regulation of gene expression without changing its sequence is provided by miRNA. Although the whole family of miRNAs can show tissue- and time-specific patterns, we believe that we can not only detect but also modulate these small polymers. First, we need to identify the miRNAs that represent a certain metabolic status, e.g., cancer. Such research has been already conducted, but depending on different study groups (different cancer stage or grade) and different methods involved (ELISA, qPCR, detection in serum, exosomes or cancer cells), it may give varying results. Some studies show alterations in seven miRNAs (miR-10b, miR-21, miR-125b, miR-145, miR-155 miR-191 and miR-382) in serum of breast cancer patients compared to healthy controls [41], while other studies reveal more than 50 different miRNAs altered in breast cancer patients [42]. Gene expression control may become a way for cancer cells to overcome therapeutic strategies, as well as an efficient way to eliminate cancer cells or make them more sensitive to therapeutic agents [43]. Thus, we performed a study that aimed to evaluate the alterations in breast cancer cell lines exposed to anticancer drugs i.e., doxorubicin or platin. First, we performed the assessment of basal levels of eleven miRNAs that are most commonly evoked when breast cancer is studied [1,2,9,10,11,12,14,16,18,19,21,44,45,46,47,48,49,50,51,52,53,54,55,56,57]. The data were demonstrated as relative miRNA levels compared to MCF7 cells, indicated as a reference cell line. Since the basal levels were extremely different (data range from 0.1 to 23 arbitrary units) and putting all the data on one graph might be misleading, we divided the results into three groups presented in three independent graphs i.e., relative miRNA levels high, low and other, relative to results observed in MCF7 cells (selected as reference cell line; Figure 2A, 2B and 2C, respectively). Noteworthy, the molecular characteristics of the three studied cell lines significantly differ, which may justify the different basal levels of the assessed miRNAs. However, it is truly difficult to tell if miRNA levels are affected by (or affect) respective features of selected cell lines. Additionally, we should not forget that all three cell lines are derived from three different people and, as commonly known, represent not only different genotypes but also heterogeneous population of cancer cells. However, one of the critical differences between studies cells is the ER/PR/HER2 receptors status. Importantly, these receptors mediate cell proliferation, growth, metabolism and other signaling pathways that, since they are related to the mechanisms affected by the studied miRNAs (Table 2), seem to justify alterations in their basal levels. Similarly, studied cells are characterized by different p53 statuses (i.e., MCF7/wt, MDA-MB-231 and SK-BR-3/mut), which is one of the key players not only in cell proliferation control and apoptosis but also in response to exposition to DNA-damaging compounds. Even with that knowledge, it is difficult to find a pattern regarding basal levels of selected miRNAs in studied cell lines.

After evaluation of the cytotoxic activity of doxorubicin and cisplatin, we performed an analysis of the alterations of target miRNAs in cells subjected to selected low-cytotoxicity concentrations of studied chemotherapeutics in MCF7, MDA-MB-231 and SK-BR-3. Although relative changes were observed in the accumulation of most of the analyzed miRNAs after drug administration, we focused on the changes that showed at least 40% change relative to control samples. Thus, essential alterations were observed in a couple of miRNAs that showed a trend in all cell lines subjected to two DNA-damaging agents.

The miRNA that was considerably altered in two of the three cell lines (MDA-MB-231 and SK-BR-3) was miR-21. It was already suggested that identification of this miRNA in serum of breast cancer patients can be used for breast cancer diagnosis at an early stage of the disease. Although it was not associated with the status of ER, PR and clinical stages [58], it was reported that miR-21 could be related to the development of Multi Drug Resistance (MDR) in breast cancer [59]. Specifically, miR-21 was shown to contribute to breast cancer proliferation and metastasis by targeting LZTFL1 [60]. As reported in colorectal cancer, an increase in miR-21 expression correlated with resistance to fluorouracil therapy due to lowered expression of the repair protein MSH2 [61]. Thus, it is possible that treatment of cells with ABC substrates may provoke alterations in one of the MDR drivers, i.e., miR-21.

It is known that miR-106 is significantly upregulated in human breast cancer, as it can enhance cell proliferation, colony-forming capacity, migration and invasion. Additionally, miR-106a overexpression significantly decreased BC cell apoptosis and sensitivity to cisplatin [62]. It was also shown that upregulation of miRNA-106a modified DNA damage response and led to the suppression of the ATM gene and formation of its protein product at nuclear foci [63]. Thus, it may be concluded that induction of this miRNA in cells treated with DOX or cis-Pt could indicate a response of the cancer cells to the DNA-damaging agent. Consequently, it might enhance the resistance of breast cancer cells to cis-Pt; although we do not know the exact mechanism, we might try to target this miRNA to attenuate the resistance effect. Importantly, mir-106a is known to be involved in DNA damage repair systems and cause sensitization of cancer cells to irradiation by targeting the 3′-UTR of ATM kinase. It was found upregulated in breast cancer cells subjected to DNA damage induction [64]. Importantly, this pathway is associated with checkpoint protein 2 (Chk2), mediating the effects of ATM on DNA damage repair mechanisms and other cellular responses that consequently halt the cell cycle (phosphorylates p53) [12]. We also showed that this miRNA was upregulated in MDA-MB-231 and SK-BR-3 cells as well as in all cell lines after cis-Pt. Lack of significant alteration of miR-106a in MCF7 cells after DOX treatment might be associated with the wild type of p53 in those cells, which has a much shorter half-life than the mutated form [65]. However, such correlation verification would require a certain signaling feedback assessment.

Another altered miRNA was miRNA-155. It was not expressed in SK-BR-3 cells, while it was significantly reduced in MCF7 and induced in MDA-MB-231 after DOX treatment, while induction was observed in both cell lines after cis-Pt treatment. In turn, the mir155-5p was downregulated in MCF7 cells after either DOX or cis-Pt treatment. Administration of DOX provoked induction of this miRNA in the other two cell lines, while cis-Pt treatment showed similar effect in MDA-MB-231 and SK-BR-3 cells. Specifically, miR-155 was found to be overexpressed in breast cancers [66]. It is also known to suppress apoptosis in MDA-MB-453 breast cancer cells by blocking caspase-3 activity [67]. It can also promote loss of genomic integrity in cancer cells by targeting genes involved in microsatellite instability and DNA repair, which strengthens the oncogenic features of this miRNA. It was also shown to decrease chemosensitivity to cisplatin in colon cancer cells and caspase-3 activity induced by cisplatin [45]. Additionally, it was found to be upregulated in the doxorubicin-resistant human lung cancer A549/DOX cell line [46]. As demonstrated previously, miR-155-5p accelerated DNA damage repair, which led to resistance to radiation of esophageal carcinoma cells [68]. However, a contrary observation was made in breast cancer, in which it was revealed that miR-155-5p decreased the efficiency of homologous recombination repair and enhanced sensitivity to radiation by targeting RAD51 directly [69]. This may be due to the different interactions of miRNA-mRNAs in different types of cancer that is also a known fact [70]. Thus, it seems that downregulation of this miRNA, which accompanied drug treatment, might be a good prognostic factor that could show high efficacy for the therapy in breast cancer. miR-155 is one of just a few miRNAs studied in the context of response to DNA-damaging drugs in breast cancer [71].

Another miRNA that was significantly altered in all three cell lines was miR-199. Specifically, miR-199b-5p was downregulated in DOX- as well as cis-Pt-treated MCF7 cells. It was upregulated in DOX-treated MDA-MB-231 cells and cis-Pt-treated SK-BR-3 cells. In turn, 199b-3p was downregulated after cis-Pt treatment in MCF7 and MDA-MB-231 cells, while in SK-BR-3, it was upregulated after DOX treatment. This particular miRNA was shown modulated in ovarian and prostate cancers, osteosarcoma and hepatocellular carcinoma but also in breast cancer. There are studies suggesting that miR-199b-5p is involved in the Notch signaling pathway in osteosarcoma and its downregulation is correlated with aggressive clinical characteristics of breast cancer [18,19]. In fact, the relative level of this miRNA was lower in the most invasive cell line studied in our work, i.e., MDA-MB-231 cells. According to the literature, downregulation of miR-199b-5p is correlated with poor prognosis for breast cancer patients [72]. Thus, modification of this miRNA may show some prognostic value in the context of breast cancer therapy monitoring. However, even if it seems to be a very sensitive marker, it may also be a very unstable marker that requires further profiling in a dynamic environment of cancer cells subjected to drugs. In previous reports, miR-199a-3pwas shown to be induced in response to DNA damage mediated by homologous repair system that suggests involvement of mTOR and c-Met [73]. Similarly, miR-199-5p/3p was shown to target DNA-damage inducible 1 homolog 2, which also implies involvement of this miRNA in response to DNA-damaging agents [74]. This miRNA is known to significantly diminish aggressive progression, including cell oxygen consumption, colony formation and mobility of breast cancer cells [21]. Variable changes of this polymer after DOX or cis-Pt treatment suggest important role of this miRNA in the response of breast cancer cells to therapy.

Another miRNA modulated after DOX or cis-Pt treatment was miR-335-3p (in SK-BR-3 cells). It is known to be associated with p53 in a positive feedback loop to drive cell cycle arrest indicating its important role in proliferation control of cancer cells [75]. The expression level of miR-335 in tissues and cells varies with cancer types, and miR-335 has been proposed as a potential biomarker for the prognosis of cancer. Besides, miR-335 may serve as an oncogene or tumor suppressor via regulating different targets or pathways in tumor initiation, development and metastasis. Furthermore, miR-335 also influences tumor microenvironment and drug sensitivity [21]. Importantly, overexpression of miR-335 was shown to increase the sensitivity of triple-negative breast cancer cells to paclitaxel, cisplatin and doxorubicin, and improve the effect of chemotherapy, as demonstrated in breast cancer patients [76]. We could not see any significant alteration in the expression of either miR-335-3p or miR-335-5p in MDA-MB-231 (triple negative) or MCF7 cells. However, it is difficult to state if alteration in the level of this miRNA after cancer cell exposure to a drug (with a decrease most noticeable in SK-BR-3 cells) contributes to the protection or toxicity mechanism. However, it was suggested that this miRNA could be a tumor suppressor and could serve as a potential therapeutic target for breast cancer treatment [77]. However, again, further studies on the mechanism involved in different cancer types and metabolic conditions or therapy regimen must be evaluated.

### 4.1. Clinical Relevance

The ultimate goal for scientific studies is providing tools for controlling biological processes to achieve the most wanted outcomes that are length and/or quality of life. This approach requires analysis of the data that may significantly contribute to the metabolism of cell and all human body. These data include genetic code as well as gene expression profiling that is covered by The Cancer Genome Atlas (TCGA) [78]. We wanted to evaluate the clinical relevance of the studied miRNAs in breast cancer data panel. Such assessment was performed using the algorithm available at oncolnc.org [40]. Although numerous studies show potential effect of target miRNAs modulation on cell survival or resistance, a general assessment of the survival relative to low or high selected miRNAs levels did not show any significant contribution of studied miRNAs expression to this parameter (Figure 4). Nine out of eleven of all studied miRNAs were found in the base but, surprisingly, no significant association of any of the miRNAs and patients’ survival was found. It may result from still low data numbers in TCGA (high and low—296 cases each group) that may not be sufficient when facing breast cancer, which is a very heterogeneous disease. Especially since, as previously reported, different roles of selected miRNAs in different breast cancer types (ER/PR/HER2 positive vs. negative) were reported [76], presented in Table 2. Although there is no significant difference in the survival time relative to the studied miRNAs’ expression, some apparent trends can be recognized, but it may require larger group studies and evaluation of other parameters, such as cancer stage or grade, p53 variant, etc., to obtain conclusive results.

### 4.2. Study Limitations

There are many studies focused on the assessment of the role of miRNA in cancer development and diagnostics. The potential of miRNA to monitor cancer response to therapy is also raised. However, serious challenges must be faced before more unequivocal conclusions are delivered. First, it must be taken into account that breast cancer (as many other cancer types) reveals high heterogeneity that is driven by many factors, including ethnically diverse backgrounds, age at diagnosis, stage at diagnosis and genetic and non-genetic alterations (including genomic, transcriptomic, proteomic and epigenetic). This diversity of tumor cells’ profiles led to distinguishing different classification levels, i.e., based on histology and expression profiles of the molecular markers; estrogen receptor (ER), progesterone receptor (PR) and the overexpression or gene amplification of human epidermal growth factor receptor 2 (HER2). According to the presence or absence of these critical receptors, specific molecular subtypes were selected: Luminal A (ER and PR-positive, HER2-negative, low Ki67), Luminal B (ER and/or PR positive, HER2-positive or high Ki67), HER2-enriched (ER and PR-negative, HER2-positive) and triple-negative (TNBC) (ER, PR, HER2-negative). They all exhibit distinct clinical outcomes and require different treatment strategies [79]. The seminal studies using gene expression profiling have further subdivided breast cancers into molecular and transcriptomic subtypes of prognostic and predictive importance. Thus, we see some limitations of our work that refers to three cell lines only that did not provide representative or coherent data.

Additionally, referring to the subject of the study, i.e., miRNA, we are aware that it is commonly known for its non-specific action and broad target profile. This, in turn, requires more advanced studies involving specific miRNA downregulation or induction to observe their specific role in affecting specific molecular pathways that control selected functional mechanisms in cancer cells. We also encountered some technical issues. As commonly acknowledged, there is no perfect reference gene for miRNA evaluation. Even the U6 reference gene is sometimes questioned as not stable enough. Another issue is drug selection and its concentration and treatment time during the study. All these factors may significantly affect the cancer cell response that is followed by the different responses of cells and, consequently, different alterations in cell metabolism and gene expression. Lastly, some experiments enable observation of early response, while others provide information regarding the prolonged effect. Altogether, it seems that the evaluation of the role of miRNAs in cancer response to therapy is based on the assessment of a very subtle and sensitive to changes parameter. Thus, it may be difficult to obtain conclusive results before reaching a broader context and evaluating samples derived from patients with different cancer types and characteristics. Without a doubt, miRNA downregulation, mRNA and protein profiling and functional studies are required that will show how all these modulations affect cancer cell metabolism and, eventually, the patient’s outcomes.

### 4.3. Potential Mechanism

Evaluation of the biological potential of selected miRNAs in breast cancer patients was performed using targetscan.org (access date: 27 February 2023) [37] and literature data (as shown in Table 2). Based on these analyses, all selected miRNAs could trigger significant effect on the expression of genes contributing to the most critical features of cancer cells i.e., proliferation, adhesion, DNA damage/repair pathways, apoptosis, autophagy, etc. (see Table 2). Identification of potential targets for selected miRNAs showed that miR-21-3p could affect numerous genes but some of them were predominantly associated with cancer homeostasis, such as DNA damage-regulated autophagy modulator 1 (DRAM1), DNA damage-inducible transcript 4-like (DDIT4L) and p53 and DNA damage-regulated 1 (PDRG1). A similar analysis also showed that miR-126-5p is associated with DDIT4L and PDRG1, but also with growth arrest and DNA damage-inducible, gamma (GADD45G). In turn, miR-335-3p levels corresponded with the expression of GADD45A and mediator of DNA-damage checkpoint 1 (MDC1). Literature data were even more abundant but mostly referred to in vitro conditions. Although the role of selected miRNAs in tumor development and response to therapy is well documented, it is still difficult to state if the observed associations are the cause or the result of alterations observed during carcinogenesis or therapeutic agent treatment. For many years, it was thought that the expression of miRNA in cancer cells was primarily reduced. Only a comparison of the miRNA profile of normal and cancer tissues showed significant overexpression of some miRNAs [80]. Depending on the function of miRNAs in the development of tumors, they are classified as suppressor miRNAs (inhibiting the expression of oncogenes or genes that induce apoptosis) and oncogenic miRNAs (activating oncogenesis or inhibiting the expression of suppressor genes) [81]. It should be emphasized that this classification is a significant simplification, because in the case of many miRNAs (e.g., miR-155), the effect of their activity depends on the total activity of regulated genes and tissue type [81]. Although miRNA levels seem very variable, we are still convinced that they can be used as a diagnostic marker and a potential target in modern anticancer therapies. However, due to the enormous number of this short polymers and no need for full complementarity to act, it may be difficult to find conclusive remarks. It seems that different conditions, including time and concentration or a type of therapeutic strategy, significantly affect the observed alteration. Nevertheless, they still seem to be a promising target that reflects not only a disease-associated modulation of the metabolism, but could also reflect the response of cancer cells to therapy that can be monitored. Consequently, targeting specific miRNAs could also be an important element of an efficient therapeutic approach. We suggest that the alterations due to drugs treatment of certain miRNA fractions depend on the breast cancer cell line characteristics. However, these preliminary results require further detailed studies in vitro and in vivo to verify their clinical potential in monitoring and therapy based on miRNAs profiling and targeting.

## Figures and Tables

**Figure 1 genes-14-00702-f001:**
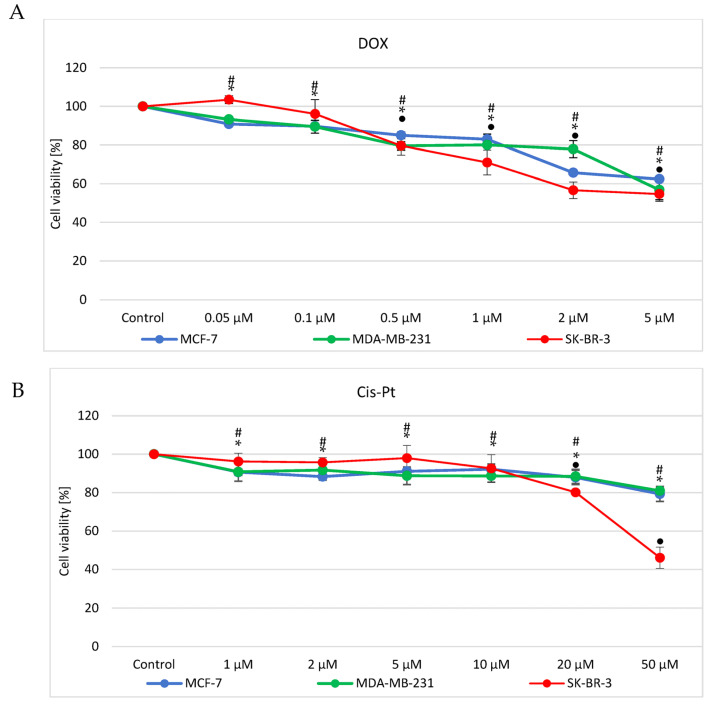
Cytotoxicity assessment and determination of the optimal range of drugs concentration i.e., doxorubicin (**A**) and cisplatin (**B**). MTT assay was performed to find optimal concentrations of doxorubicin and cisplatin for evaluation of miRNAs alterations as a response. Cells were treated for 24 h, followed by colorimetric assessment. All experiments were performed in triplicates (each with six technical repeats). Alterations in survival are relative to control, untreated cells. *, •, # *p* < 0.05.

**Figure 2 genes-14-00702-f002:**
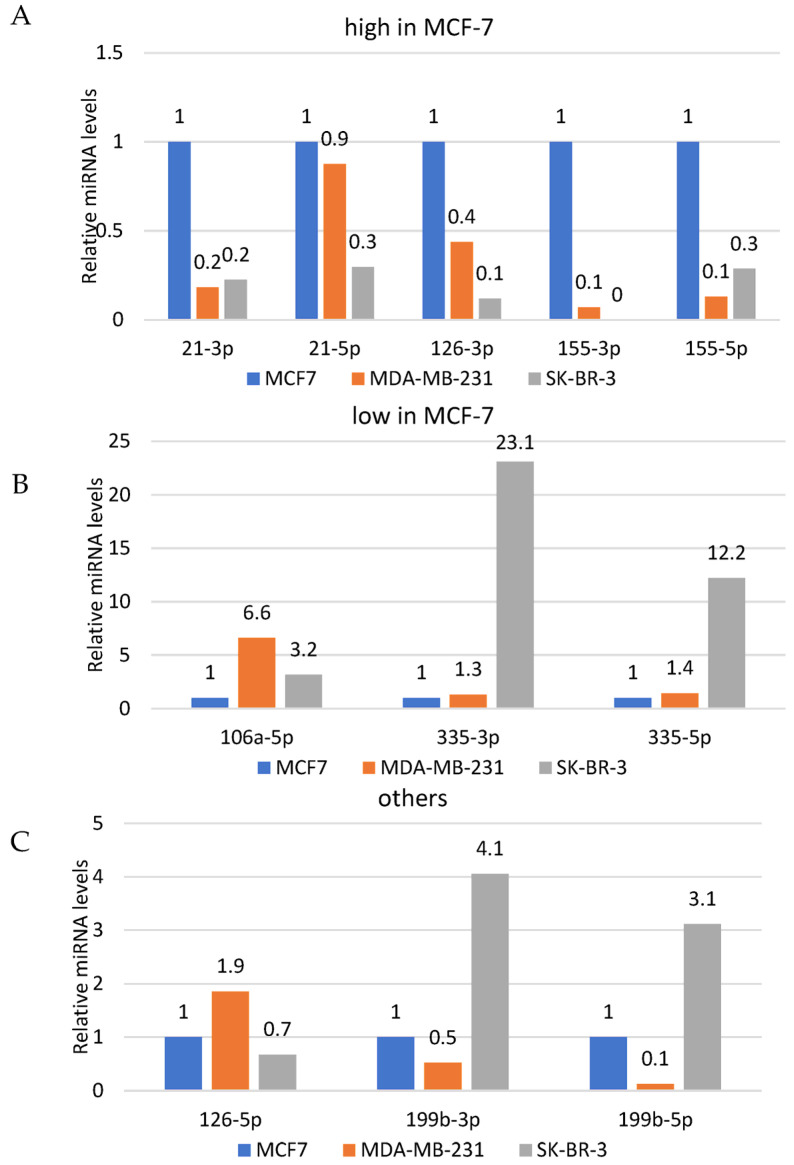
Quantitative assessment of the basal levels of selected miRNAs using qPCR in studied cell lines. miRNAs expression levels were grouped according to their pattern of expression for better representation. The data for all three cell lines (i.e., (**A**), MCF7; (**B**), MDA-MB-231; (**C**), SK-Br-3) were divided into three categories i.e., low, high and others, relative to MCF7 cells, selected as the reference cell line (value “1”).

**Figure 3 genes-14-00702-f003:**
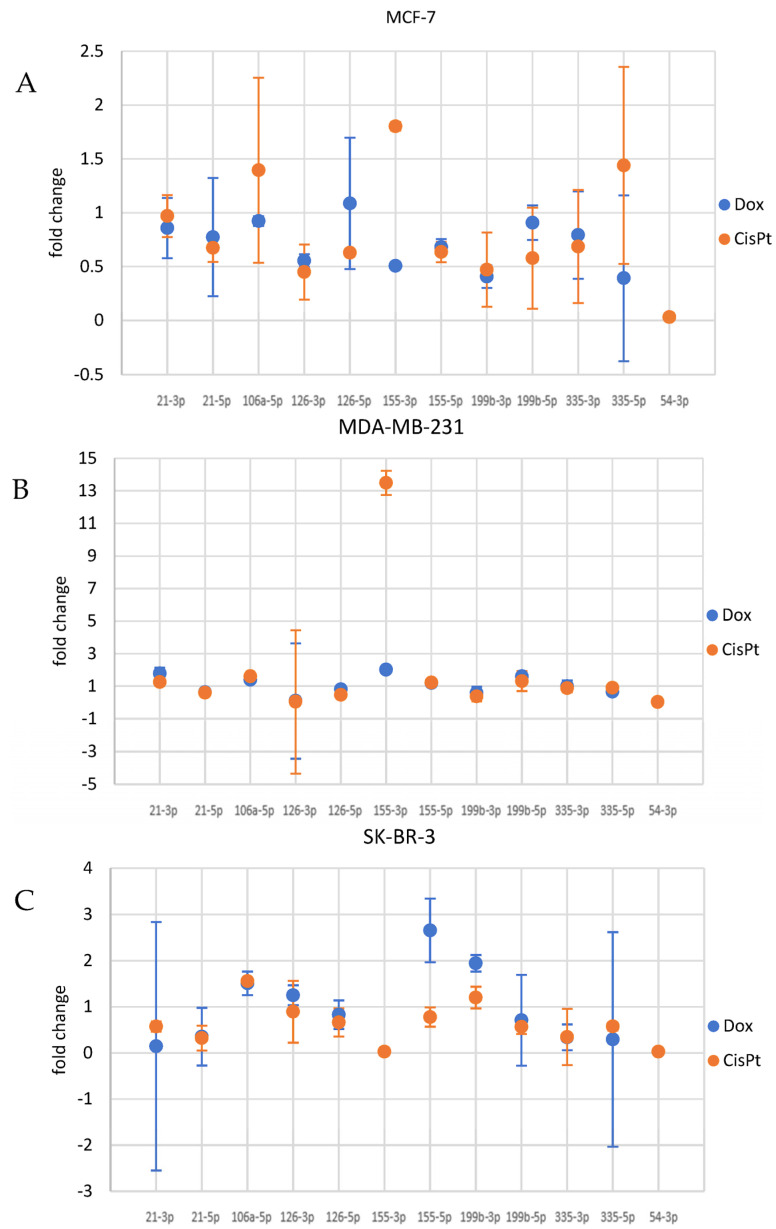
Evaluation of cancer cells response to doxorubicin or cisplatin measured by miRNA levels alterations. All three cell lines ((**A**), MCF7; (**B**), MDA−MB−231; (**C**), SK−BR−3) were subjected to selected miRNAs assessment after treatment with DOX (0.1 μM) or cis-Pt (10 μM) for 24 h. Experiments were performed in triplicates using RT-qPCR assessment. The values represent fold changes, relative to MCF7 cells (used as reference). Values higher than “1” mean induction, while values lower than “1” mean reduction of selected miRNAs levels.

**Figure 4 genes-14-00702-f004:**
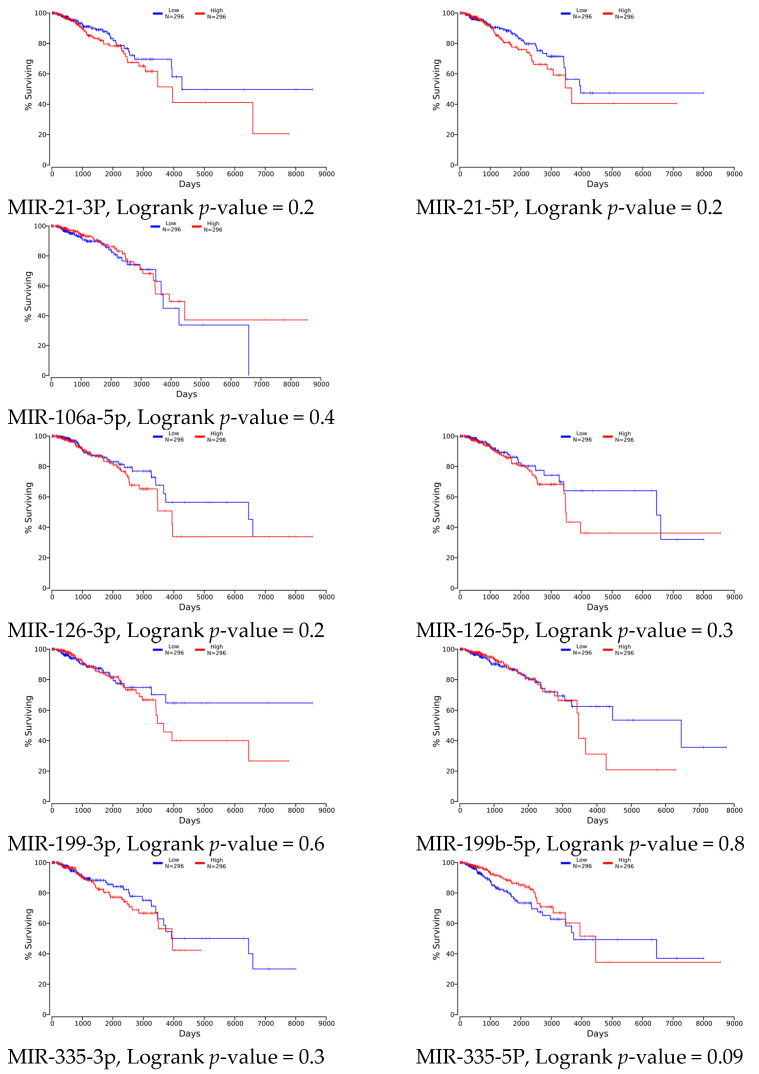
Association between studied miRNAs levels and patients’ survival estimated with the use of miRNA-seq data submitted in TCGA. Lower and upper percentile were set at 30; high and low expression was estimated in 296 cases each group; data analyzed in [40].

**Table 2 genes-14-00702-t002:** IC_50_ values determination for doxorubicin or cisplatin after MCF7, MDA-MB-231 or Sk-BR3 cells treatment for 24 h. IC_50_ values were calculated using CompuSyn program (three biological repeats, six technical repeats each, were performed).

	Drug	IC_50_ [µM]	
Cell Line		DOX	Cis-Pt
MCF7		27.4 ± 0.9	>50
MDA-MB-231		12.9 ± 1.8	>50
SK-BR-3		5.8 ± 0.7	44.3 ± 2.6

## Data Availability

All relevant data are included in the manuscript.

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
