# Peer review of "Doxorubicin and Cisplatin Modulate miR-21, miR-106, miR-126, miR-155 and miR-199 Levels in MCF7, MDA-MB-231 and SK-BR-3 Cells That Makes Them Potential Elements of the DNA-Damaging Drug Treatment Response Monitoring in Breast Cancer Cells—A Preliminary Study"

_genes, 2023, doi:10.3390/genes14030702_

Round 1
Reviewer 1 Report
The manuscript describes changes in expression of miRNAs in drug treated breast cancer cells. The authors need to validate their findings with more experiments. The manuscript gives a general impression of miRNA expression. Please find the detailed comments below -
1. The authors need to calculate the IC50 value for each drug-cell line. Tabulate this information and state clearly the concentrations that will be used for assaying miRNAs.
2. Figure 3 shows miRNA expression in drug treated cells. It would be nice to see miRNA expression levels in all three cell lines together in a given treatment. This would tell which miRNA was most impacted by drug treatment in a given setting.
3. The authors need to choose few miRNAs that were significantly affected by treatment and perform their knockout/overexpression studies to completely validate their finding.
4. The authors can show the expression of the selected miRNAs using TCGA data to prove the significance of these miRNAs in breast cancer.
Overall the study is interesting, but needs validation for conclusion.
Author Response
First, we would like to thank the reviewers for their kind comments and constructive criticism. Please find addressed all comments below and in the text. All changes in the manuscript are made in a track changes mode. As the studies were performed in vitro, we also slightly altered the title of the manuscript to emphasize that fact (current version: Doxorubicin and cisplatin modulate miR-106, miR-155 and miR-199 levels in MCF7, MDA-MB-231 and SK-BR-3 cells that makes them potential elements of DNA-damaging drugs treatment response monitoring in breast cancer cells).
Reviewer 1
Regarding general comments:
The manuscript describes changes in expression of miRNAs in drug treated breast cancer cells. The authors need to validate their findings with more experiments. The manuscript gives a general impression of miRNA expression. Please find the detailed comments below.
We agree that our results show just a preliminary aspect of the role of miRNA in the monitoring of cancer cells’ response to therapeutic agents. And certainly, some more advanced studies like miRNA knockout/overexpression in target cells would definitely broaden the picture and address the mechanistic aspect. It is the plan for further steps of our studies that require the involvement of much more time and resources. We are aware that the mited) selected cell lines, as well as drugs and treatment conditions, cannot provide the full picture of the role of miRNAs in drug-treated cells but we believe we show some potential of these particles. The cell type (breast cancer) selection was based on worldwide statistics and our previous experiments and main area of interest. Additionally, after literature data research, we selected certain miRNA and justification for this selection can be found in the Results section as well as in the discussion part of the current version of the manuscript.
- The authors need to calculate the IC50 value for each drug-cell line. Tabulate this information and state clearly the concentrations that will be used for assaying miRNAs.
IC50 values were calculated using CompuSyn program and are now shown in table 1. (three biological repeats, and 6 technical repeats each were performed). Concentrations used for miRNA alterations assessment were chosen from the low-cytotoxic range that was supposed to show specific metabolic effects without cytotoxicity-associated response. Thus, being far from IC50 values, the applied concentrations of the drugs refer to the viability curves. Importantly, these concentrations were also tested in longer time intervals (48 and 72h) but in these conditions, they appeared highly cytotoxic which implied some not specific effects that were not within our scope.
- Figure 3 shows miRNA expression in drug treated cells. It would be nice to see miRNA expression levels in all three cell lines together in a given treatment. This would tell which miRNA was most impacted by drug treatment in a given setting.
It is presented now in the new Figure 3. We modified the graphs by removing the control/reference values (“1”) and showed just fold change of the miRNAs alterations. Thus we show four graphs that represent studied miRNAs expression levels in all three cell lines together in a given treatment that tells which miRNA was most impacted by drug treatment in a given setting. It is presented now in the new Figure 3.
- The authors need to choose few miRNAs that were significantly affected by treatment and perform their knockout/overexpression studies to completely validate their finding.
Yes, that is a perfect suggestion but impossible to do within a decent amount of time (not to mention about 10 days that we were given for a revision). However, this suggestion seems particularly precious and will be verified in the nearest future.
- The authors can show the expression of the selected miRNAs using TCGA data to prove the significance of these miRNAs in breast cancer.
We have tried to perform TCGA analysis (gepia2, Xena browser) but since data on the selected and altered miRNA (106, 155, and 199) are scant in the literature, the analysis was not possible. We base our concept and evaluation on the most current literature data that was included in the manuscript.
Overall the study is interesting, but needs validation for conclusion.
We appreciate the comments and look forward to proceeding with some more studies to obtain more conclusive results. Respective comments were addressed and all feasible amendments were performed.

Reviewer 2 Report
Summary: The manuscript entitled "Doxorubicin and cisplatin modulate miR-106, miR-155 and miR-199 levels in MCF7, MDA-MB-231 and Sk-Br-3 cells that makes them potential elements of therapy response monitoring in breast cancer" aimed at exploring the change in expression levels of miRNAs in breast cancer cell lines in response to two chemotherapeutic agents: cisplatin and doxorubicin.
The authors mainly depended on 3 breast cancer cell lines (MCF7, MDA-MB-231, and Sk-br-3), that belong to different breast cancer subtypes, to tackle the question. The authors focused on 11 miRNAs to assess based on literature review. Although, the article tackles a valid and interesting question in oncology practice and cancer research, several major concerns exist for the current version as follows:
A) Experimental execution and data presentation:
1) Figure 1 shows results of survival assay after a dose-response curve for Dox and Cis. It is not clear whether this experiment was replicated 3 times biologically or technically. The tightness of the error bars infer that these are technical replicates (done at the same time). Authors should clarify.
2) The premise is to define the pattern of expression of these miRNAs in terms of therapy response. Authors should clarify the rationale of choosing a short time interval (24 hrs). Are they interested in early response rather than prolonged? And why? The same point calls for caution regarding the chosen title for the article. Such title implies rigorous follow up (different timings, etc) rather than exploratory studies that the authors nicely performed.
3) In Figure 2, authors compare the three cell lines in terms of different miRNAs expression level. This is an interesting piece of data that was not sufficiently discussed. Another suggestion would be grouping the miRNAs according to their pattern of expression for better representation. For instance: i) category 1: high in MCF7 (21-3p, 21-5p, 126-3p, 155-3p, 155-5p) ii) category 2: low in MCF7 (106a-5p, 335-3p, 335-5p) iii) others: 126-5p, 199b-3p, 199b-5p.
4) In the assessment of miRNAs expression levels in response to treatment, it is not clear why the authors chose two doses from each drug that didn’t give different results in survival analysis (Figure 1). Authors should clarify the rationale.
5) Figure 3 is difficult to follow: (i) the figure might benefit from splitting the information on different panels or even expanding the data onto multiple figures to make it easier for the reader to follow. For instance: two different panels for each cell line, one for dox and another for cis. The “c” condition can also be omitted since it is already set at 1. (ii) the figure is not showing any variation between replicates, the three replicate fold change from control should be represented. This is critical for concluding from such techniques.
B) Organizational:
1) In the introduction, the authors do not cite relevant references supporting the candidacy of miRNAs as prognostic tools generally. What would be the advantage of depending on miRNAs levels rather than (or complementary to) currently used tests in the clinic?
2) The section 1.1. Target miRNAs selection (line 59) might belong better in the results section where the authors actually started their efforts by reviewing different miRNAs in cancer research. Another suggestion would be presenting these conclusions in a table where the miRNA name, cancer lineage/cell line, and net effect can be presented.
3) Actual mechanisms of drug resistance/response, which is the main focus of the manuscript, are not discussed or introduced.
Author Response
Reviewer 2
First, we would like to thank the reviewers for their kind comments and constructive criticism. Please find addressed all comments below and in the text. All changes in the manuscript are made in a track changes mode. As the studies were performed in vitro, we also slightly altered the title of the manuscript to emphasize that fact (current version: Doxorubicin and cisplatin modulate miR-106, miR-155 and miR-199 levels in MCF7, MDA-MB-231 and SK-BR-3 cells that makes them potential elements of DNA-damaging drugs treatment response monitoring in breast cancer cells).
Summary: The manuscript entitled "Doxorubicin and cisplatin modulate miR-106, miR-155 and miR-199 levels in MCF7, MDA-MB-231 and Sk-Br-3 cells that makes them potential elements of therapy response monitoring in breast cancer" aimed at exploring the change in expression levels of miRNAs in breast cancer cell lines in response to two chemotherapeutic agents: cisplatin and doxorubicin.
The authors mainly depended on 3 breast cancer cell lines (MCF7, MDA-MB-231, and Sk-br-3), that belong to different breast cancer subtypes, to tackle the question. The authors focused on 11 miRNAs to assess based on literature review. Although, the article tackles a valid and interesting question in oncology practice and cancer research, several major concerns exist for the current version as follows:
- A) Experimental execution and data presentation:
1) Figure 1 shows results of survival assay after a dose-response curve for Dox and Cis. It is not clear whether this experiment was replicated 3 times biologically or technically. The tightness of the error bars infer that these are technical replicates (done at the same time). Authors should clarify.
The results are representative of three independent replicates performed week by week. Each biological repeat is an average of six technical repeats. We do realize these SD bars seem slight but since the MTT test was performed by very experienced researchers (ET and NL) there is no doubt they are solid and remain in concordance with previous data from our laboratory.
2) The premise is to define the pattern of expression of these miRNAs in terms of therapy response. Authors should clarify the rationale of choosing a short time interval (24 hrs). Are they interested in early response rather than prolonged? And why? The same point calls for caution regarding the chosen title for the article. Such title implies rigorous follow up (different timings, etc) rather than exploratory studies that the authors nicely performed.
We agree that our results show just a preliminary aspect of the role of miRNA in the monitoring of cancer cells’ response to therapeutic agents. And certainly, some more advanced studies like miRNA downregulation/miRNA transfection in target cells would definitely broaden the picture and raise the mechanistic aspect. It is the plan for further steps of our studies that requires the involvement of much more time and resources. We are aware that the (limited) selected cell lines, as well as drugs and treatment conditions, cannot provide the full picture of the role of miRNAs in drug-treated cells but we believe we show some potential of these particles. The cell type (breast cancer) selection was based on worldwide statistics and our previous experiments and the main area of interest. Additionally, after literature data research, we selected certain miRNAs, and justification for this selection can be found in the Results section as well as in the discussion part of the current version of the manuscript.
Regarding the time interval that was evaluated i.e. 24h – it was based on our previous experience and numerous cytotoxicity assays as well as more mechanistic assessment of apoptotic pathways. We concluded that longer incubation time (48 and 72h MTT tests were performed, showing more than 40% viability decrease in all three cell lines at the lowest concentrations of both studied drugs; data not included) would provoke cytotoxic effect that might not reflect the mechanistic aspect of the specificity of the cancer cells respond to the treatment. Also, according to our previous studies, many genes are regulated within 24h after DOX or Cis-Pt. Similarly, numerous papers indicate 24h time interval as a sufficient time to observe significant alterations in miRNA levels after cancer cells treatment (e.g. http://www.aginganddisease.org/EN/10.14336/AD.2016.1109; https://www.nature.com/articles/nbt.1543; https://www.ncbi.nlm.nih.gov/pmc/articles/PMC6662485/; https://www.nature.com/articles/s41598-017-03689-7). For this reason, we were interested in a rapid response that was supposed to be specific. Although many examples for different time intervals can be found in the literature. However, we decided to verify our hypothesis concerning the monitory potential of miRNAs in a 24h time interval. Consequently, we also slightly altered the title of the manuscript which is now:
Doxorubicin and cisplatin modulate miR-106, miR-155, and miR-199 levels in MCF7, MDA-MB-231, and SK-BR-3 cells that makes them potential elements of DNA-damaging drugs treatment response monitoring in breast cancer cells
3) In Figure 2, authors compare the three cell lines in terms of different miRNAs expression level. This is an interesting piece of data that was not sufficiently discussed. Another suggestion would be grouping the miRNAs according to their pattern of expression for better representation. For instance: i) category 1: high in MCF7 (21-3p, 21-5p, 126-3p, 155-3p, 155-5p) ii) category 2: low in MCF7 (106a-5p, 335-3p, 335-5p) iii) others: 126-5p, 199b-3p, 199b-5p.
Figure 2 was modified but most importantly, the data were discussed and miRNAs expression levels were grouped according to their pattern of expression for better representation. The data for all three cell lines were divided into three categories i.e. low, high, and others.
4) In the assessment of miRNAs expression levels in response to treatment, it is not clear why the authors chose two doses from each drug that didn’t give different results in survival analysis (Figure 1). Authors should clarify the rationale.
As mentioned above, we used some literature premises and performed 24h treatment and miRNAs evaluation. Regarding two values, we decided to make research in the low-cytotoxic range of concentration. Additionally, in longer time treatment experiments we observed that these low concentrations eventually induced a cell survival decrease (48 and 72h, data not included).
5) Figure 3 is difficult to follow: (i) the figure might benefit from splitting the information on different panels or even expanding the data onto multiple figures to make it easier for the reader to follow. For instance: two different panels for each cell line, one for dox and another for cis. The “c” condition can also be omitted since it is already set at 1. (ii) the figure is not showing any variation between replicates, the three replicate fold change from control should be represented. This is critical for concluding from such techniques.
As mentioned above, the organization of Figure 3 was changed and in the current version, it shows just the fold change of the miRNA alterations in a given treatment but for all three cell lines that tell which miRNA was most impacted by drug treatment in a given setting. Thus we show four graphs that represent studied miRNAs expression levels in all three cell lines together in a given treatment.
Regarding variations - as mentioned in the manuscript, qPCR was performed with pooled samples as commonly acknowledged in such experiments. Precisely, all biological treatment experiments were performed in triplicates (by pooling the RNA samples) followed by RT-PCR, and qPCR assessment, giving mean target miRNAs levels.
- B) Organizational:
1) In the introduction, the authors do not cite relevant references supporting the candidacy of miRNAs as prognostic tools generally. What would be the advantage of depending on miRNAs levels rather than (or complementary to) currently used tests in the clinic?
The introduction section was amended and references were updated accordingly. Briefly, we suggest, based on evoked references, that miRNAs could serve as early biomarkers for the evaluation of drug efficacy and drug safety. Such an approach was proposed not only for breast cancer but also for other diseases including multiple sclerosis, bipolar disorder, diabetes, and others. Additionally, we discuss the advantages of miRNAs assessment in pharmacogenomics as they play an important role as modulators of pharmacology-related gene expression that provides a fast, low-invasive methods for high specificity and sensitivity monitoring of drug metabolism.
2) The section 1.1. Target miRNAs selection (line 59) might belong better in the results section where the authors actually started their efforts by reviewing different miRNAs in cancer research. Another suggestion would be presenting these conclusions in a table where the miRNA name, cancer lineage/cell line, and net effect can be presented.
Target selection was transferred to the results section and the justification was summarized in Table 2.
3) Actual mechanisms of drug resistance/response, which is the main focus of the manuscript, are not discussed or introduced.
First, the used drugs characteristics were provided in the Materials and Methods section. The resistance aspect was elaborated on in the introduction section and discussed accordingly. We raised the aspect of many variables that could contribute to the response of cancer cells to a certain drug, including dose-dependent effects (in our case concentration-dependent) as well as exposure time. Additionally, we emphasized the role of p53 in the drug-response mechanism, and, importantly, the studied cell lines represent breast cancer cells with different statuses of this protein, i.e. - MCF7 (wt), MDA-MB-231 and SK-BR-3 (mut).

Round 2
Reviewer 1 Report
I would like to go with my previous comment that the manuscript describes changes in expression of miRNAs in drug treated breast cancer cells. The authors need to validate their findings with more experiments. The manuscript gives a general impression of miRNA expression. The authors have revised the manuscript but not addressed the previous comments completely. Please find the detailed comments below.
1. The authors need to include a housekeeping gene like U6 or RNU-49 in the real time PCR experiment.
- Please prepare a panel for Figure 2.
- The authors state, " SK-BR-3 were missing the miR-155-3p". The authors can rephrase to "did not show expression".
- The authors need to choose few miRNAs that were significantly affected by treatment and perform their knockout/overexpression studies to completely validate their finding.
5. The authors can utilize the readcounts/raw miRNA-seq data submitted in public repositry to the expression of the selected miRNAs using TCGA data to prove the significance of these miRNAs in breast cancer.
6. The authors need to come up with a mechanism of drug using the miRNA expression profile.
7. The study is open ended and needs substantial experimental validation.
Author Response
We believe, we addressed all the comments in the present form of the manuscript.
Rev 1
I would like to go with my previous comment that the manuscript describes changes in expression of miRNAs in drug treated breast cancer cells. The authors need to validate their findings with more experiments. The manuscript gives a general impression of miRNA expression. The authors have revised the manuscript but not addressed the previous comments completely. Please find the detailed comments below.
- The authors need to include a housekeeping gene like U6 or RNU-49 in the real time PCR experiment.
Replicates for one selected concentration of applied drugs were provided and are shown relative to U6 expression (Figure 2 and 3), as recommended by the reviewer and justified by respective reference [Lou, G.; Ma, N.; Xu, Y.; Jiang, L.; Yang, J.; Wang, C.; Jiao, Y.; Gao, X. Differential distribution of U6 (RNU6–1) expression in human carcinoma tissues demonstrates the requirement for caution in the internal control gene selection for microRNA quantification. Int. J. Mol. Med. 2015, 36, 1400–1408].
Additionally, for better representation of the results, a scatter plot was applied with +/- SD values in Figure 3.
- Please prepare a panel for Figure 2.
Smaller graphs are provided that enables the formation of a panel but further decreasing the size may make the figure difficult to read.
- The authors state, " SK-BR-3 were missing the miR-155-3p". The authors can rephrase to "did not show expression".
It was corrected.
- The authors need to choose few miRNAs that were significantly affected by treatment and perform their knockout/overexpression studies to completely validate their finding.
We agree and elaborate on the response below (Ad 7).
Additionally, information and the potential role of selected (significantly altered) miRNAs in breast cancer was elaborated, supported by TCGA analysis.
- The authors can utilize the readcounts/raw miRNA-seq data submitted in public repository to the expression of the selected miRNAs using TCGA data to prove the significance of these miRNAs in breast cancer.
A paragraph “Clinical significance” in the discussion section was added that comprises the TCGA data analysis regarding miR-21-3p, miR-21-5p, miR-21-5p, miR-106a-5p, miR-126-3p, miR-126-5p, miR-199-3p, miR-199b-5p, miR-335-3p, and miR-335-5p. However, no significant association between the expression of any studied miRNA and patients’ survival was revealed (http://www.oncolnc.org/, lower and upper percentile at 30, comparing low and high expression of selected miRNAs, 296 cases in each group).
- The authors need to come up with a mechanism of drug using the miRNA expression profile.
A section “Potential mechanism“ was added based on https://www.targetscan.org/ and potential target genes evaluation. As elaborated, the observed alterations in selected miRNAs after drug treatment suggest their involvement in DNA-damage response and growth arrest/control in cancer cells (miRNAs: 21-3p, 126-5p, and 335-3p).
- The study is open ended and needs substantial experimental validation.
Yes, we agree that the study is open.
Just to emphasize, our goal was to assess the alterations in miRNA levels in subcytotoxic concentration of DNA-damaging drugs.
Importantly, it is one of just a few studies showing the possibility of using miRNA as a monitoring tool for drug efficacy in breast cancer.
However:
- since miRNA does not have to be fully complementary to act
- the response mechanism depends on cell characteristics
- the response mechanism depends on drug type and concentration
it may be difficult to obtain conclusive results.
Even blocking single miRNAs will not provide unequivocal data. Thus further studies will require miRNAs downregulation, as well as mRNA and protein profiling and functional studies that will show how all these modulations affect cancer cell metabolism and, eventually, the patients’ outcomes.
So we are following this way, but we are still not ready to address all these steps.

Reviewer 2 Report
In an appreciable effort, the authors addressed the majority of the comments. However, a remaining concern is to actually represent the qPCR data from the three replicates individually (i.e. scatter plot) rather than the mean. Again, this is a misleading representation. This should be a trivial change for the authors.
Author Response
We believe, we addressed all the comments in the present form of the manuscript.
Rev 2
In an appreciable effort, the authors addressed the majority of the comments. However, a remaining concern is to actually represent the qPCR data from the three replicates individually (i.e. scatter plot) rather than the mean. Again, this is a misleading representation. This should be a trivial change for the authors.
Replicates for one selected concentration of applied drugs were provided and are shown relative to U6 expression. Additionally, for a better representation of the results, a scatter plot was applied with +/- SD values in Figure 3.
